# Effect of the Use of Metronome Feedback on the Quality of Pediatric Cardiopulmonary Resuscitation

**DOI:** 10.3390/ijerph18158087

**Published:** 2021-07-30

**Authors:** Dongjun Yang, Wongyu Lee, Jehyeok Oh

**Affiliations:** Department of Emergency Medicine, Chung-Ang University College of Medicine, 84 Heukseok-ro, Dongjak-gu, Seoul 06974, Korea; tomtony1000@gmail.com (D.Y.); victory9294@gmail.com (W.L.)

**Keywords:** cardiopulmonary resuscitation, feedback, manikins, pediatrics

## Abstract

Although the use of audio feedback with devices such as metronomes during cardiopulmonary resuscitation (CPR) is a simple method for improving CPR quality, its effect on the quality of pediatric CPR has not been adequately evaluated. In this study, 64 healthcare providers performed CPR (with one- and two-handed chest compression (OHCC and THCC, respectively)) on a pediatric resuscitation manikin (Resusci Junior QCPR), with and without audio feedback using a metronome (110 beats/min). CPR was performed on the floor, with a compression-to-ventilation ratio of 30:2. For both OHCC and THCC, the rate of achievement of an adequate compression rate during CPR was significantly higher when performed with metronome feedback than that without metronome feedback (CPR with vs. without feedback: 100.0% (99.0, 100.0) vs. 94.0% (69.0, 99.0), *p* < 0.001, for OHCC, and 100.0% (98.5, 100.0) vs. 91.0% (34.5, 98.5), *p* < 0.001, for THCC). However, the rate of achievement of adequate compression depth during the CPR performed was significantly higher without metronome feedback than that with metronome feedback (CPR with vs. without feedback: 95.0% (23.5, 99.5) vs. 98.5% (77.5, 100.0), *p* = 0.004, for OHCC, and 99.0% (95.5, 100.0) vs. 100.0% (99.0, 100.0), *p* = 0.003, for THCC). Although metronome feedback during pediatric CPR could increase the rate of achievement of adequate compression rates, it could cause decreased compression depth.

## 1. Introduction

For improvement in the survival rate among pediatric patients having cardiac arrest, the performance of high-quality cardiopulmonary resuscitation (CPR) has been emphasized in recent CPR guidelines [1,2,3]. Aspects of high-quality CPR include adequate chest compression depth (CCD), an adequate chest compression rate (CCR), complete recoil between chest compressions, minimal interruption of chest compressions, and the avoidance of hyperventilation [1]. It has been reported that the quality of CPR performed on the spot is low; thus, the use of various types of feedback devices has been attempted [4,5,6]. Consequently, for the improvement in the quality of CPR, the use of feedback devices during the provision of resuscitation training has been recommended in CPR guidelines [7,8,9]. Of the different types of feedback provided through various feedback mechanisms, audio feedback provided using a metronome is a simple tool for maintaining an adequate CCR. It has been confirmed in previous studies that through the performance of chest compressions with sound and audio tone feedback, adequate CCRs could be maintained [10,11,12,13]. However, the observation of decreased CCD with the use of sound and audio tone feedback has been reported in several studies [11,13]. Additionally, in one study, it was found that changes in the performance of CPR (for people having out-of-hospital cardiac arrest (OHCA)) due to the provision of real-time visual and audible feedback by a monitor-defibrillator were not associated with improvements in clinical outcomes, such as the return of spontaneous circulation or survival rates [14]. Therefore, if there is an unavailability of feedback devices that provide directive feedback regarding CCRs, CCD, chest compression release, and hand position, the provision of audio feedback with music or metronomes is recommended in CPR guidelines [7,8,9].

With respect to pediatric resuscitation, in one study in which researchers evaluated whether the use of a metronome improves CCR and CCD during CPR performed on a pediatric manikin, the researchers found that, in an in-hospital cardiac-arrest setting in which the two-handed chest-compression (THCC) technique is used, with the provision of audio feedback produced by a metronome, the mean percentage of compressions delivered within an adequate rate could be improved without decreased CCD [10]. To the best of our knowledge, the effect of metronome feedback on the quality of CPR performed on pediatric patients having cardiac arrest in an out-of-hospital setting has not been evaluated yet. Additionally, the use of one of two different chest-compression techniques (THCC or one-handed chest compression (OHCC), Figure 1) based on the thoracic size of a child and the hand span of the individual who is performing CPR on the child is recommended in CPR guidelines [1,2,3]. Therefore, in the present study, we evaluated the effect of the use of metronome feedback on the quality of CPR performed with both THCC and OHCC for out-of-hospital pediatric cardiac arrest. We hypothesized that the use of metronome feedback would lead to decreased CCD.

## 2. Materials and Methods

### 2.1. Study Design

This was a prospective, randomized, crossover trial. Participants were assigned to one of two groups, group A and B, using randomization lists created with random-number sequences obtained with a web-based program, to obtain six permuted blocks with the initials of each group: “A” or “B”. The participants assigned to group A first performed CPR for 2 min with metronome feedback (test 1); then, they performed CPR for 2 min without metronome feedback (test 2). In contrast, the participants assigned to group B first performed CPR for 2 min without metronome feedback, followed by the performance of CPR for 2 min with metronome feedback. To avoid participant fatigue, we provided the participants with a rest period of 30 min between the tests (Figure 2). For all participants, two rounds of tests were carried out (each participant performed CPR four times). In the first round, the THCC technique was used for chest compression. In the second round, the OHCC technique was used for chest compression. We provided over one day of rest between two rounds to avoid participant fatigue.

### 2.2. Study Setting and Population

This study was conducted in the emergency department of a university hospital between March 2021 and June 2021 using a simulated pediatric out-of-hospital cardiac arrest model. We used the Resusci Junior QCPR pediatric manikin (Laerdal, Stavanger, Norway) as a substitute for a child having cardiac arrest. The manikin was the size of a 5-year-old child. The manikin was laid on the hard floor of the emergency department in a supine position. Audio feedback for the performance of chest compressions was provided using an electronic metronome (SMT-1000, Samick Musical Instruments Co., Ltd., Chungcheongbuk-do, Korea), which produced a low-pitched sound at a rate of 110 beats/min. Healthcare providers who were certified to provide basic life support participated in the study after providing written informed consent. The exclusion criteria were inability to perform CPR because of a recent hand or arm injury and refusal to participate in the study. Ultimately, 64 healthcare providers were recruited.

### 2.3. Sample Size Calculation

CCD was considered the primary outcome variable; based on this consideration, the sample size for this study was calculated. For a two-sided test of significance, the significance level was set at 0.05, with a power of 80%. In a previous study, the average CCD achieved without audio tone feedback was found to be 39.3 mm (standard deviation (SD): 9.5 mm), and that achieved with audio tone feedback was found to be 35.8 mm (SD: 8.2 mm) [13]. Therefore, we set the allowable difference as 3.5 mm. The minimum number of participants that would be required for each group was calculated using a web program (sample size calculator: two crossover-sample means) and was determined to be 32, considering a loss rate of 10% [15].

### 2.4. Study Protocol

After recruiting the participants, we collected basic information (data regarding gender, age, job position, and handedness) about them before the performance of the tests. In each test, CPR was performed for 2 min by a single participant (rescuer), with a compression-to-ventilation ratio of 30:2. Mouth-to-mouth ventilation was performed using a face shield. The positions of the participants and manikin were standardized as follows: each right-handed participant was placed on the right side of the manikin and each left-handed participant was placed on the left side of the manikin. For OHCC, the dominant hand was used for chest compressions, and the non-dominant hand was used to maintain the manikin’s head-tilt position during chest compressions. All participants performed CPR a total of four different times (in two rounds) because we tested two different chest-compression techniques (the THCC and OHCC techniques were used in the first and second rounds, respectively). Data on chest-compression parameters, such as CCD or CCRs, were collected in real-time by a sensor embedded in the manikin and were extracted using a device (SimPad SkillReporter; Laerdal, Stavanger, Norway). The screen of the device was hidden and not visible to either the participants or the researcher.

### 2.5. Outcome Variables

The primary outcome variable was average CCD (mm). The secondary outcome variables included the rate of use of correct hand position (percentage), total number of compressions, the rate of achievement of adequate CCD (percentage), the rate of achievement of complete recoil (percentage), average CCR (compressions/min), the rate of achievement of an adequate CCR (percentage), hands-off time (sec), total ventilation (number), and average ventilation volume (mL).

### 2.6. Statistical Analysis

All statistical analyses were performed using IBM^®^ SPSS^®^ Statistics 26.0 (IBM, Armonk, New York, NY, USA). Values for continuous variables have been presented as mean ± SD or median (interquartile range), according to the normality of data distributions; values for categorical variables have been presented as frequencies and percentages. Data were analyzed using the Shapiro–Wilk test to evaluate the normality of distributions. For variables with normally distributed data, the paired *t*-test was used to compare variables; otherwise, the Wilcoxon signed-rank test was used to compare variables. The *p*-values < 0.05 were considered statistically significant.

## 3. Results

Sixty-four healthcare providers were recruited and assigned randomly to the two groups (Figure 2). None of the participants were excluded.

### 3.1. Participant Characteristics

Most of the study participants were men (53/64, 82.8%) and right-handed (60/64, 93.8%). The mean age of the participants was 27.7 years (SD: 2.9 years). With respect to job positions, 59.4% (38/64) of the participants were resident physicians, and 40.6% (26/64) were interns.

### 3.2. Comparisons of Variable Values for CPR Performed with and without Metronome Feedback

The rate of achievement of an adequate CCR during CPR performed with metronome feedback was significantly higher than that during CPR performed without metronome feedback; however, for both chest-compression techniques, the rate of achievement of adequate CCD during CPR performed with metronome feedback was significantly lower than that during CPR performed without metronome feedback (Table 1). For both chest-compression techniques, the average CCD achieved during CPR performed without metronome feedback was significantly greater than that achieved during CPR performed with metronome feedback (Figure 3). With respect to CPR performed with THCC, the rate of achievement of complete recoil during CPR performed with metronome feedback was lower than that during CPR performed without metronome feedback (Table 1); however, in this regard, for CPR performed with OHCC, there was no significant difference between CPR performed with and without metronome feedback. For CPR performed with THCC, the average CCR during CPR performed with metronome feedback was also lower than that during CPR performed without metronome feedback (Figure 4); however, with respect to average CCRs, for CPR performed with OHCC, there was no significant difference between CPR performed with and without metronome feedback. With respect to parameters associated with ventilation, regardless of the technique of chest compression, there were no differences between CPR performed with and without metronome feedback.

## 4. Discussion

The results of the present study were simple and clear. The results supported our hypothesis that metronome feedback (audio feedback) during CPR, provided using a metronome, would lead to decreased CCD. Similar results have been observed in the previous studies [11,13]. However, previous studies involved the evaluation of CPR performed for adults having cardiac arrest in out-of-hospital settings during which the THCC technique was used. Therefore, to the best of our knowledge, the present study is the first study in which the unfavorable effect of the use of audio feedback on the quality of CPR performed for pediatric cardiac arrest has been confirmed.

It is unclear why there is decreased CCD with audio feedback. The authors of a previous study suggested that decreased CCD was caused due to the complexity of performing chest compression while receiving feedback [13]. Individuals who do not receive audio feedback during CPR performance can easily perform chest compression according to known CPR techniques, whereas individuals receiving audio feedback must combine the known CPR techniques with the auditory perception of feedback received in real-time during CPR. The process might require the individual performing CPR to multitask in a short period of time, leading to an increased workload; this could simultaneously lead to decreased CCD.

Although the differences in the average CCD values (during CPR performed with and without metronome feedback) were within 3 mm for both OHCC CPR and THCC CPR, it is possible that decreased CCD could have decreased the adequacy of chest compression. Additionally, we cannot determine the risk-benefit ratio associated with the benefit of controlling CCRs and the risk of decreased CCD (through a risk-benefit analysis).

The effect of the use of feedback during pediatric CPR had been evaluated in several past studies [10,16,17]. In only one of these previous studies, the effect of the use of a metronome during CPR was evaluated [10]; however, decreased CCD, an adverse effect of the use of audio feedback that was observed in the present study, was not observed in the previous study. With respect to study settings, there are several differences between the previous study and the present study. In the previous study, audio feedback was provided using an audible metronome that sounded at a rate of 100 times per minute [10]; however, in the present study, due to the CCR recommended in CPR guidelines (100–120 compressions/min), audio feedback was provided using a metronome that sounded at a rate of 110 times per minute [1,2,3]. It is possible that this comparatively higher rate of sound production for audio feedback could have affected CCD. Second, the study setting of the previous study was an in-hospital setting [10]. As a result, in the previous study, the cardiac-arrest simulator manikin was placed on a board on a standard hospital bed and mattress. With respect to environments in which CPR is performed, CPR performed in in-hospital settings differs from that performed in out-of-hospital settings. Significantly, in in-hospital settings, the posture of rescuers is affected by the relationship between beds and rescuers [18,19,20,21]. In addition, hospital beds and mattresses could affect the quality of CPR performed [22,23,24,25]. The standard posture and position for a rescuer performing CPR is based on the occurrence of cardiac arrest in out-of-hospital settings, in which patients lie on hard floors. Therefore, these differences could have affected the occurrence of decreased CCD in the present study and the lack thereof in the previous study.

The advantages of audio feedback are clear. With audio feedback during CPR, the CCR can be precisely controlled with a predetermined speed. The data obtained through the present study also indicate that audio feedback has a positive effect on the quality of CPR.

Additionally, irrespective of the provision of audio feedback, the CCD achieved during CPR performed using the THCC technique was found to be greater than that achieved during CPR performed using the OHCC technique (*p* < 0.001). It is well known that increased CCD is achieved with THCC [26]; compared to CPR with OHCC, CPR with THCC produces significantly higher pressure, which causes increased CCD [27].

This study has several limitations. First, we only recruited healthcare providers who were certified to provide basic life support. Therefore, our results cannot be applied to other types of rescuers. Moreover, in our study, CPR was performed on the floor (out-of-hospital setting). Therefore, different results may be obtained for CPR performed on a manikin or individual on a bed (in-hospital CPR setting). Second, since the results of the present study were obtained using a simulator manikin, the results may not be identical to results obtained through studies in which CPR is performed on humans. In addition, the differences regarding environment between simulated pediatric OHCA situations and real-life situations in which the performance of CPR is required might have affected the motivation and performance of the participants. Third, the present study evaluated the performance of the CPR for only 2 min. Therefore, if the duration of CPR exceeds 2 min, different results might be obtained.

## 5. Conclusions

The use of audio feedback produced by a metronome during CPR performed on a pediatric patient having cardiac arrest in an out-of-hospital setting could improve the rate of achievement of adequate CCR during CPR, regardless of the chest-compression technique used. However, with respect to both OHCC and THCC, decreased CCD was observed when CPR performance was guided using sound produced by a metronome. Therefore, for CPR performed on pediatric patients having cardiac arrest in out-of-hospital settings, we should not use only audio feedback produced by a metronome; in order to ensure that adequate CCD is achieved along with an adequate CCR, in addition to the use of metronome feedback, the simultaneous use of a feedback device that could be used to correct CCD and achieve adequate CCD is needed.

## Figures and Tables

**Figure 1 ijerph-18-08087-f001:**
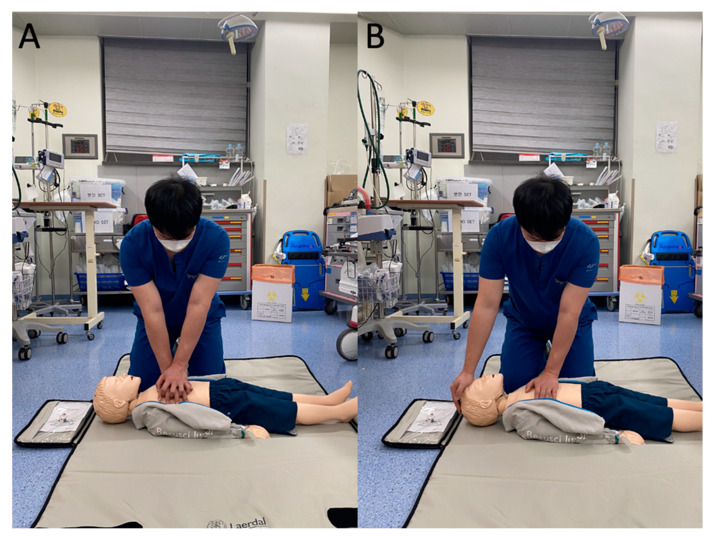
Two different chest-compression techniques recommended for pediatric cardiopulmonary resuscitation. (**A**) Two-handed chest compression; (**B**) one-handed chest compression.

**Figure 2 ijerph-18-08087-f002:**
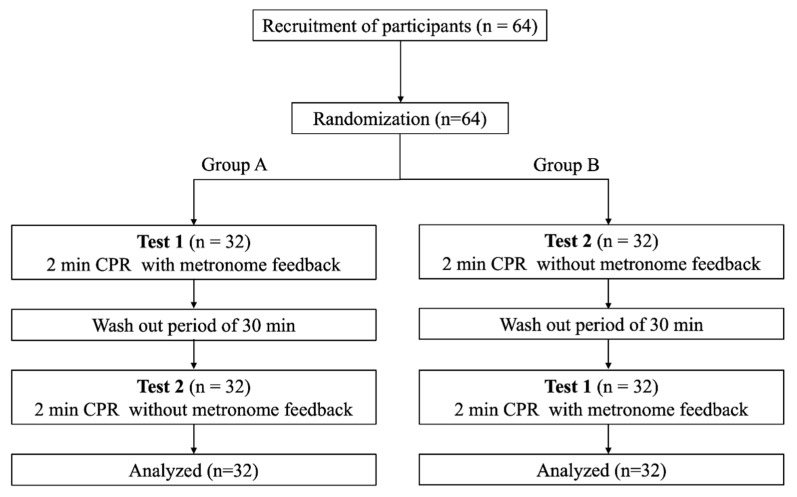
Study design. CPR: cardiopulmonary resuscitation.

**Figure 3 ijerph-18-08087-f003:**
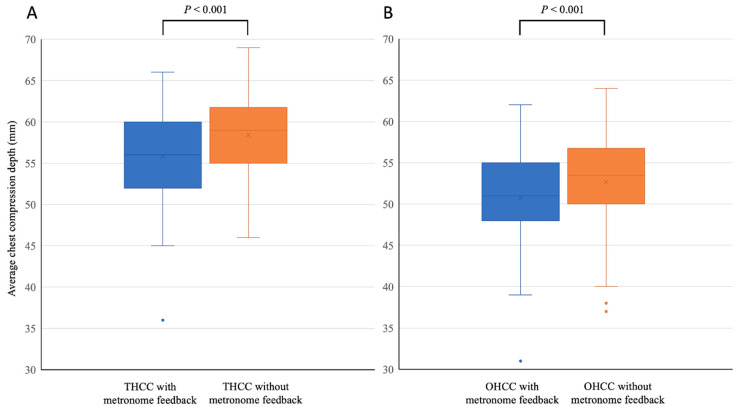
Comparisons of average chest compression depth during cardiopulmonary resuscitation performed with and without audio feedback provided using a metronome. (**A**) Pediatric cardiopulmonary resuscitation performed using the two-handed chest-compression (THCC) technique; (**B**) pediatric cardiopulmonary resuscitation performed using the one-handed chest-compression (OHCC) technique.

**Figure 4 ijerph-18-08087-f004:**
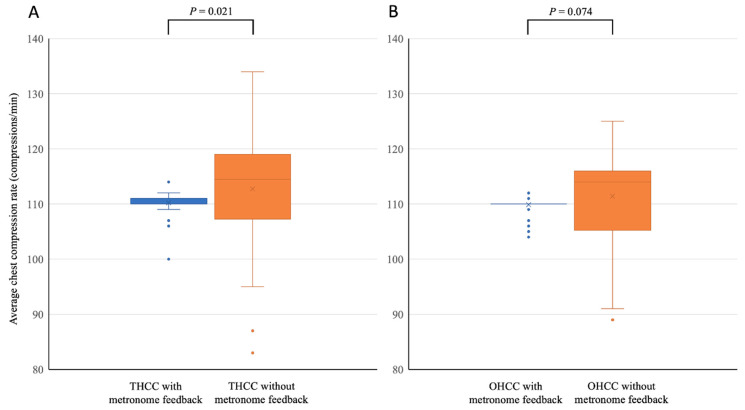
Comparisons of average chest compression rates during cardiopulmonary resuscitation performed with and without audio feedback provided using a metronome. (**A**) Pediatric cardiopulmonary resuscitation performed using the two-handed chest-compression (THCC) technique; (**B**) pediatric cardiopulmonary resuscitation performed using the one-handed chest-compression (OHCC) technique.

**Table 1 ijerph-18-08087-t001:** Comparison of variable values for cardiopulmonary resuscitation performed with and without metronome feedback.

Variables	THCC with Metronome Feedback	THCC without Metronome Feedback	*p* Value
Rate of use of correct hand position (%)	100.0 (100.0, 100.0)	100.0 (100.0, 100.0)	0.955
Total compressions (number)	160.2 ± 8.8	161.1 ± 15.5	0.596
Average CCD (mm)	55.8 ± 5.6	58.4 ± 4.9	**<0.001**
Adequate CCD achievement rate (%)	99.0 (95.5, 100.0)	100.0 (99.0, 100.0)	**0.003**
Complete-recoil achievement rate (%)	96.0 (68.0, 100.0)	97.5 (86.0, 100.0)	**0.006**
Average rate (compressions/min)	110.0 (110.0, 111.0)	114.5 (107.5, 119.0)	**0.021**
Adequate CCR achievement rate (%)	100.0 (98.5, 100.0)	91.0 (34.5, 98.5)	**<0.001**
Hands-off time (sec)	8.0 (7.0, 9.0)	8.0 (7.5, 9.0)	0.050
Total ventilation (number)	10.0 (9.5, 10.0)	10.0 (9.0, 10.0)	0.796
Average ventilation volume (mL)	159.1 ± 40.3	158.7 ± 45.7	0.891
**Variables**	**OHCC with Metronome Feedback**	**OHCC without Metronome Feedback**	***p* Value**
Rate of use of correct hand position (%)	100.0 (100.0, 100.0)	100.0 (100.0, 100.0)	0.223
Total compressions (number)	161.6 ± 8.7	161.0 ± 13.8	0.698
Average CCD (mm)	51.0 (48.0, 55.0)	53.5 (50.0, 56.5)	**<0.001**
Adequate CCD achievement rate (%)	95.0 (23.5, 99.5)	98.5 (77.5, 100.0)	**0.004**
Complete-recoil achievement rate (%)	100.0 (97.5, 100.0)	100.0 (97.5, 100.0)	0.674
Average rate (compressions/min)	110.0 (110.0, 110.0)	114.0 (105.5, 116.0)	0.074
Adequate CCR achievement rate (%)	100.0 (99.0, 100.0)	94.0 (69.0, 99.0)	**<0.001**
Hands-off time (sec)	8.0 (7.0, 9.0)	8.0 (7.0, 9.0)	**0.015**
Total ventilation (number)	10.0 (9.5, 10.0)	10.0 (9.0, 10.0)	0.563
Average ventilation volume (mL)	164.0 (133.0, 185.5)	162.0 (122.5, 184.5)	0.256

*p*-values < 0.05 are presented in bold; CCD: chest compression depth; CCR: chest compression rate; OHCC: one-handed chest compression; THCC: two-handed chest compression.

## Data Availability

The data presented in this study are available on request from the corresponding author (J.O.).

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
