# Peer review of "Effect of the Use of Metronome Feedback on the Quality of Pediatric Cardiopulmonary Resuscitation"

_ijerph, 2021, doi:10.3390/ijerph18158087_

Round 1

Reviewer 1 Report

Dear authors,

I hope to contribute constructively to your research with the following review.

The manuscript investigates whether the use of metronomes during pediatric CPR with one or two hands brings benefits to the quality of the maneuvers. Performs a prospective, randomized, and crossover, study; considering feedback from a realistic simulation manikin.

Broad comments

The authors suspect that this work is unpublished. However, the references presented are insufficient for such suspicion. A systematic review of the literature will increase its sturdiness to consideration.

Specific comments

Line 25-26: All keywords must be in the abstract.

Line 56-59 and 189-191: Scores the Broad comment.

Line 228-230: This statement should be further detailed, considering which parameter the current guidelines consider most important in CPR (CCD or CCR).

Sincerely,

Reviewer.

Author Response

  1. The authors suspect that this work is unpublished. However, the references presented are insufficient for such suspicion. A systematic review of the literature will increase its sturdiness to consideration.

Answer) Thank you for your valuable comment. Audio-visual feedback for cardiopulmonary resuscitation (CPR) had been evaluated several times and even a systematic review and meta-analysis already had been published in 2013 (PMID: 24361457). However, most of the studies had been conducted using adult model. In addition, feedback effect for one-handed chest compression technique had not been evaluated yet. In the area of pediatric resuscitation, only few studies had been tried to evaluate the effect of feedback on CPR (Child model: PMID: 26459645, 30700565, 25531167, 30231037; Infant model: PMID: 29944894, 31164375, 29084067, 23571117, 30850868). Therefore, more studies should be needed to conduct a systematic review on the effect of feedback in the area of pediatric resuscitation. We could not find the study which evaluate the effect of feedback on the paediatric CPR using one-handed chest compression technique. In addition, we also could not find the study which evaluate the effect of metronome guidance on the pediatric CPR in an out-of-hospital setting. Therefore, we described the sentence “To the best of our knowledge, the effect of metronome guidance on the quality of CPR performed on pediatric patients having cardiac arrest in an out-of-hospital setting has not been evaluated yet.”

  1. Line 25-26: All keywords must be in the abstract.

Answer) Thank you for your valuable comment. We have revised the keywords considering the review’s comment (Keywords: cardiopulmonary resuscitation; feedback; manikins; pediatrics). In addition, the word of “guidance” had been replaced with “feedback” in all manuscript including title and figures because the word of “guidance” is not a MeSH terminology.

  1. Line 56-59 and 189-191: Scores the Broad comment.

Answer) I’m sorry. I cannot understand the meaning of the reviewer’s comment. Please let us know what point should be enhanced in our manuscript.

  1. Line 228-230: This statement should be further detailed, considering which parameter the current guidelines consider most important in CPR (CCD or CCR).

Answer) Both of adequate chest compression depth (CCD) and chest compression rate (CCR) are key components of the high-quality cardiopulmonary resuscitation (CPR) (PMID: 25252721, 25565457). However, we don’t know which parameter is more important between the CCD and CCR considering the patient’s survival. Therefore, the recent resuscitation guideline emphasizes both CCD and CCR for the conditions of high-quality CPR (PMID: 33081529).

Reviewer 2 Report

Experimental manikin study assessing the value of a metronome on the quality of cardiopulmonary resuscitation in a simulated paediatric cardiac arrest setting. The authors conclude that the rate of adequate chest compression rate improves but the chest compression depth may diminish using a metronome

Overall, well conducted study and relevant findings 

One additional major limitation should be noted: The testing lasted only two minutes. The findings may have differed more with a 10-20+ minutes study period.

Also, some findings are statistically relevant but their clinical relevance can be discussed, e.g. 99% vs. 100% adequate CCD achievement rate

Author Response

  1. The testing lasted only two minutes. The findings may have differed more with a 10-20+ minutes study period.

Answer) Thank you for your valuable comment. We have added it in the limitation section (Third, the present study evaluated the performance of the CPR for only 2 minutes. Therefore, if the duration of CPR exceeding 2 minutes, different results might be obtained).

  1. Some findings are statistically relevant, but their clinical relevance can be discussed, e.g. 99% vs. 100% adequate CCD achievement rate.

Answer) Thank you for your comments. Adequate chest compression depth (CCD) achievement rate is calculated as following method. When the measured CCD is deeper than 5 cm, the chest compression is regarded as achieving adequate CCD. If all of the chest compressions conducted by a certain study participant are deeper than 5 cm, the adequate CCD achievement rate of the study participant is calculated as 100%. The value of 99.0% in the THCC with metronome feedback group is a median value of the adequate CCD achievement rate in all participants (N=64). Although the difference in the adequate CCD achievement rate between THCC with metronome feedback group and THCC without metronome feedback group was only 1%, the actual difference in the average CCD increased as 2.6 mm. Actually, we don’t know whether this difference (only 2.6 mm difference in the CCD) could affect the clinical outcome (such as survival rate) or not. However, we can predict that the CCD could decrease when the audio feedback will be used during CPR. The purpose of our study is raising the concern, that is adverse effect of audio feedback in pediatric resuscitation. We think that further study especially clinical trials should be needed to confirm the effect of audio feedback during pediatric resuscitation on the clinical outcome. Again, thank you for your comment.

Reviewer 3 Report

The study seems interesting to me and provides the originality of being carried out in infant CPR.

The results obtained are as expected according to studies in adults. It is well designed and the document is well prepared. It is well justified and described.

Perhaps I find the results section a bit confusing, and the discussion contains parts that are difficult to read. I miss a little more comparison to previous studies.

However, in view of the results obtained, I do not agree with the statement in lines 228-229 that audio guidance has a positive effect on the quality of CPR, because the CCD is also a criterion of quality that seems worse; this result should be better described in the discussion.

Congratulations

Author Response

  1. In view of the results obtained, I do not agree with the statement in lines 228-229 that audio guidance has a positive effect on the quality of CPR, because the CCD is also a criterion of quality that seems worse; this result should be better described in the discussion.

Answer) Thank you for your careful evaluation of our manuscript. Although audio feedback resulted in decrease of the chest compression depth, it’s ability of guiding the chest compression rate at a certain target was confirmed several times in the previous studies. Therefore, the recent resuscitation guideline recommends the use of tonal guidance including music or metronome in the absence of feedback device that provide directive feedback on compression rate, depth, release, and hand position simultaneously (PMID: 33773831). However, there were limited evidence of tonal guidance in the area of pediatric resuscitation. We hope that our results could add the evidence of tonal guidance in the area of pediatric resuscitation. Again, thank you for your comment.